# A Health Risk Assessment of Lead and Other Metals in Pharmaceutical Herbal Products and Dietary Supplements Containing *Ginkgo biloba* in the Mexico City Metropolitan Area

**DOI:** 10.3390/ijerph18168285

**Published:** 2021-08-05

**Authors:** Patricia Rojas, Elizabeth Ruiz-Sánchez, Camilo Ríos, Ángel Ruiz-Chow, Aldo A. Reséndiz-Albor

**Affiliations:** 1Laboratory of Neurotoxicology, National Institute of Neurology and Neurosurgery “Manuel Velasco Suárez”, SS, Av. Insurgentes sur No. 3877, Mexico City 14269, Mexico; ruizruse@yahoo.com.mx; 2Department of Neurochemistry, National Institute of Neurology and Neurosurgery “Manuel Velasco Suárez”, SS, Av. Insurgentes sur No. 3877, Mexico City 14269, Mexico; camrios@yahoo.com.mx; 3Neuropsychiatry Unit, National Institute of Neurology and Neurosurgery “Manuel Velasco Suárez”, SS, Av. Insurgentes sur No. 3877, Mexico City 14269, Mexico; angel.ruiz@innn.edu.mx; 4Mucosal Immunity Laboratory, Research and Graduate Section, Instituto Politécnico Nacional, Superior School of Medicine, Plan de San Luis Esq. Salvador Díaz Mirón s/n, C.P., Mexico City 11340, Mexico; alrealdo@yahoo.com.mx

**Keywords:** *Ginkgo biloba*, heavy metal content, pharmaceutical herbal products, dietary supplements

## Abstract

The use of the medicinal plant *Ginkgo biloba* has increased worldwide. However, *G. biloba* is capable of assimilating both essential and toxic metals, and the ingestion of contaminated products can cause damage to health. The aim of this study was to investigate the safety of manganese (Mn), copper (Cu), lead (Pb), arsenic (As), and cadmium (Cd) in 26 items containing *Ginkgo biloba* (pharmaceutical herbal products, dietary supplements, and traditional herbal remedies) purchased in the metropolitan area of Mexico City. Metal analysis was performed using a graphite furnace atomic absorption spectrometer. All of the products were contaminated with Pb, 54% of them with As, and 81% with Cd. The lowest values of Pb, As, and Cd were detected in pharmaceutical herbal products > dietary supplements > traditional herbal remedies. The daily intake dose (DID) of pharmaceutical herbal products was within the established limits for the five metals. Dietary supplements and traditional herbal remedies exceeded the DID limits for Pb. The hazard quotients estimation and non-carcinogenic cumulative hazard estimation index for Mn, As, and Cd indicated no human health risk. Our results suggest that products containing *G. biloba* for sale in Mexico are not a health risk.

## 1. Introduction

*Ginkgo biloba*—a Chinese plant also known as ginkgo or maidenhair tree—has been used in healing treatments for thousands of years. In traditional Chinese medicine, the seeds have been used to treat bronchitis, cough, intestinal infections, enuresis, alcohol abuse, and skin infections [1]. In particular, *Ginkgo biloba* leaves are processed to obtain extracts for the treatment of cognitive disorders (memory loss and concentration)—mainly in ageing people—and circulatory disorders such as ischemic heart disease, vascular tinnitus, thromboses, vascular vertigo, and arrhythmias [2]. In this sense, two standardized extracts of *G. biloba* leaves have been scientifically documented: EGb761, and LI 1370 [3]. These diverse therapeutic actions for human health make *G. biloba* one of the most popular medicinal plants, with high demand worldwide. The extracts of *G. biloba* leaves are sold as herbal medicine, dietary supplements, and traditional herbal remedies.

The World Health Organization (WHO) defines herbal medicine as plant-derived materials or preparations used for therapeutic purposes or other health aids in humans [4]. Furthermore, the WHO defines a dietary supplement as materials indicated to complement the diet (other than tobacco), including herbs, botanical products, vitamins, amino acids, minerals, and dietary substances [4].

However, herbal products including *G. biloba* leaf extract are sold as dietary supplements in the United States according to the Dietary Supplement Health and Education Act of 1994 [5], and are commercially available without evidence of effectiveness and safety being required. In addition, the approval of the US Food and Drug Administration (FDA) is not needed. In contrast, in Mexico, France, and Germany, for example, *G. biloba* leaf extract (standardized herbal extract) is regulated as a herbal medicine by health authorities, like other prescription pharmaceutical medications that require registration based on scientific studies supporting their use.

Dietary supplements are freely accessible, with lax regulation, low costs, and consumers believing them to be equivalent in outcome to pharmaceutical products. Dietary supplements containing *G. biloba* are among the most widely sold herbal supplements on the US market [6], with an estimated global market value of USD 1.59 billion in 2018 [7]. *G. biloba* leaf extracts (non-standardized extracts) are available on the US market, as well as worldwide, as herbal supplements through networks and retail stores in the US [8], with worldwide sales of up to USD 1.2 billion [9,10]. However, due to the high demand for the purchase of dietary supplements of herbal origin, worldwide production has increased, but in several cases without the appropriate sanitary regulations. Therefore, the content of these products can differ from one manufacturer to another, and can even be polluted during the production process by heavy metals, affecting the quality of the product.

The main sources of heavy metal contamination are related to agricultural activities that include (1) where the plant was cultivated (near metal mines, foundry factories, or roadways); (2) contamination of soil, (3) irrigation with untreated water, and (4) excessive use of lead-based pesticides, arsenic-based insecticides, or fertilizers containing cadmium, which are still used in various countries [11,12,13]. Other sources include industrial processes, such as improper manufacturing conditions, or the transportation of products in open-bed vehicles, allowing the accumulation of heavy metals from the emissions of petrol engines [11,12,13]. In particular, the lead (Pb) content in the soil of urbanized areas is very high due to human activity—for example, paint containing this toxic metal used in buildings and houses, before restrictions, along with atmospheric emissions of Pb from gasoline used in motor vehicles through the years prior to its elimination by law. This caused the accumulation of lead in urban soil over the years [14]. Accordingly, the herbal products on the market are likely polluted with heavy metals incorporated in the plants, which are finally introduced into the human body, affecting the heart, liver, lungs, and kidneys, as well as the circulatory and central nervous systems [15,16,17].

Heavy metal contamination of herbal supplements is well known; for example, cadmium (Cd) and lead (Pb) are toxic at low concentrations [18]. Cd is associated with impaired kidney function, osteoporosis, anemia, reproductive deficits, anosmia, emphysema, and chronic rhinitis [18,19,20]. Chronic exposure to Cd also plays a very relevant role in the development of pathologies associated with the cardiovascular system in adults [21], as well as in mortality [22]. Pb induces behavioral abnormalities, cognitive impairment, visual dysfunction, and raised blood pressure leading to cardiovascular pathology, as well as renal tumors [18,19,20]. Arsenic (As) exposure causes dermatological abnormalities, peripheral vascular disease, and hypertension, while chronic exposure is related to cancer in the bladder, lungs, kidneys, skin, and liver [18,19,20]. Additionally, Cd and Pb have shown carcinogenic potential [18,19,20], and Pb can cross the placental barrier, affecting the fetus [18,19,20].

Moreover, other metals known as essential trace elements, which are required in small amounts, play an important role in the physiological and metabolic functions of the body—among them copper (Cu) [23] and manganese (Mn) [24]. Nevertheless, high intake or deficiencies of these can be harmful to human health [25].

On the other hand, there is little information on concentrations of metals in commercial preparations—especially in herbal products—in Mexico. Consequently, the quality and safety of these products should be considered and evaluated because of concern for public health.

The aim of the current study was to assess the concentrations of trace elements (Cu and Mn) and toxic heavy metals (Cd, Pb, and As) in pharmaceutical herbal medicines, dietary supplements, and traditional herbal remedies containing *G. biloba* available in the Mexico City metropolitan area, in order to determine whether their content is within the acceptable standards. Therefore, it is important to estimate the level of metal pollution in order to evaluate the quality and safety of these *G. biloba* products for consumers. 

## 2. Materials and Methods

### 2.1. Chemicals and Reagents

All solutions were prepared with ultrapure water obtained from an Elix^®^ Essential 3 UV water purification system (Millipore, Bedford, MA, USA). Nitric acid 65% Suprapur® and certified metal standard solution (GFAA mixed standards) were purchased from PerkinElmer (Norwalk, CT, USA).

### 2.2. Sample Collection

All commercial products containing *Ginkgo biloba* were purchased from pharmacies, health food stores, supermarkets, and local markets in the Mexico City metropolitan area and analyzed for Mn, Cu, Pb, As, and Cd. The herbal products were physically presented in liquid (tincture) or solid (capsules, tablets, and dried leaves) form. They were chosen and bought according to popularity and availability between October 2019 and February 2020. A description of the products analyzed is shown in Table 1.

### 2.3. Cleaning Material

All laboratory materials—both glassware and plastic—used for the analysis were cleaned with laboratory detergent, washed several times with cold tap water, soaked overnight in 3% nitric acid solution, and rinsed with deionized water [26]. This was done to remove any traces of metal contamination.

### 2.4. Sample Processing and Metal Analysis

Twenty-six samples containing *G. biloba*—ten products labeled as pharmaceutical herbal medicines, thirteen as dietary supplements, and three as traditional herbal remedies (leaves obtained from markets)—are described in Table 1. All products were coded and analyzed before their expiration dates. The products presented as solids were homogenized dry using a porcelain mortar and pestle (Thomas Scientific, Swedesboro, NJ, USA) and weighed. A sample of 300 mg was taken for solid products, and 300 µL for liquids, for metal analysis. In the case of leaves, an infusion was prepared according to the producer’s instructions, and 1 mL of the solution was used for the study. Then, samples were subjected to digestion with HNO_3_ Suprapur® (Merck, Darmstadt, Germany) in metal-free polypropylene tubes using 1.5 mL and 1.2 mL of the acid for solid and liquid products, respectively. In particular, 1 mL of HNO_3_ 5% Suprapur^®^ was added to the infusions. The organic material was digested with nitric acid, and then samples were stored overnight at room temperature. Finally, after total digestion, the solution that remained of each sample was diluted to analyze Mn, Pb, Cd, As, and Cu content using a PerkinElmer AA600 graphite furnace atomic absorption spectrometer coupled to an AS 800 autosampler (PerkinElmer, Norwalk, CT, USA) [27]. The equipment was provided with a specific light source for each metal to be analyzed.

The calibration curves for each metal were prepared using commercially available certified standard solutions of known concentrations for Mn, Cu, Pb, As, and Cd (GFAA mixed standard, PerkinElmer). These curves were obtained on the same day as the analysis, and used to quantify the concentration in each sample. The samples were assayed in duplicate.

### 2.5. Calculation of Daily Intake Dose of Mn, Cu, Pb, As, and Cd

The daily intake dose (DID) of each metal analyzed was calculated in the pharmaceutical herbal products according to the manufacturer´s recommended dosage written on the product, or as recommended by a healthcare professional. In the case of the herbal dietary supplement products, the calculation of the maximum daily dose followed the manufacturer’s written instructions. For traditional herbal remedies, the dose recommended by herb sellers was used for the calculation.
DID = Metal concentration (µg) × recommended intake dosage (per day)(1)

The daily intake doses were compared to recommended metal limits of the American Herbal Products Association [28], Canada Natural and Non-Prescription Health Products [29], and US California Proposition 65 [30].

### 2.6. Non-Carcinogenic Health Risk Assessment

#### 2.6.1. Risk Calculation for Human Health

Non-carcinogenic risk estimate analysis of harmful metals due to ingestion of herbal products containing *Ginkgo biloba* was calculated according to a hazard quotient (HQ) for As, Cd, and Mn. The estimation was performed according to the guidelines of the United States Environmental Protection Agency (US EPA, 2011) [31].

The estimated daily intake (EDI) for each metal was estimated according to the manufacturer- or healthcare-professional-recommended dosage per day for the average body weight of an adult person of 65 kg [32].
EDI (mg/kg/day) = DID/ Body weight (kg)(2)

The calculation of the HQ was carried out using the following equation:HQ = EDI/RfD(3)

The reference dose (RfD) is defined as an estimation of oral exposure per day by the human population (including susceptible groups) without a significant risk of harmful effects in the course of a lifetime [32]. The RfD values according to the US EPA were available for three metals: Mn (1.4 × 10^−1^ mg/kg/day, equal to 140 µg/kg/day); inorganic As (3 × 10^−4^ mg/kg/day, equal to 0.3 µg/kg/day); and Cd (5 × 10^−4^ mg/kg/day, equal to 0.5 µg/kg/day) [32]. There is no available RfD for Cu or Pb by the US EPA; therefore, they were not included in the analysis. The HQ shows a potential health risk at a value equal or greater than 1.

#### 2.6.2. Analysis of Hazard Indicator

The hazard indicator (HI) of the cumulative risk associated with the analyzed metals was calculated using the sum of the HQs for Mn, As, and Cd [33] for each product.
HI = HQ (Mn) + HQ (As) + HQ (Cd)(4)

An HI value equal to or less than 1 indicates that exposure to these metals is not expected to cause adverse non-carcinogenic effects. However, an HI value greater than 1, according to the US EPA, cannot be suggested as a statistical probability that adverse health effects will occur. 

### 2.7. Statistical Analysis

Descriptive statistics were calculated by using Excel for Microsoft 365 MSO (Microsoft Corporation, Redmond, WA, USA). Statistical analysis, including Mann–Whitney U test, was performed using RStudio free software [34].

## 3. Results

The samples of 26 products containing *Ginkgo biloba* were analyzed. The products studied were produced in Mexico (92.3%), the USA (3.85%), and Spain (3.85%), according to manufacturers. The products sampled included pharmaceutical herbal products (38.5%), dietary supplements (50%), and traditional herbal remedies (11.5%).

The analysis showed that Mn, Cu, and Pb were present in 100% of products, while As was detected in 54% of tested items (4% traditional herbal remedies, 15% pharmaceutical herbal products, and 35% dietary supplements). Cd was found in 81% of items (4% traditional herbal remedies, 27% of pharmaceutical herbal products, and 50% dietary supplements) (Table 2).

### 3.1. Daily Intake Dose (DID)

The daily intake doses (DIDs; µg/day) of Mn, Cu, Pb, As, and Cd in pharmaceutical herbal products, dietary supplements, and traditional herbal remedies containing *Ginkgo biloba* are shown in Table 2. We reported that Mn—an essential metal—was detected in all products sampled, and the daily intake was between 0.046 µg/day—belonging to a pharmaceutical herbal drug (product code P9)—and 192.164 µg/day (dietary supplement, product code S12), and the acceptable range for the Mexican population is 2300 µg/day [35]. Therefore, we found that all products were within the recommended limits. The pharmaceutical herbal products showed the lowest DID of Mn when compared to dietary supplements (Table 3; *p =* 0.001), and also compared to traditional herbal remedies (Table 3; *p =* 0.01), which displayed the highest DID.

Cu, an essential metal, was identified in all analyzed products, and the calculated daily intake was from 0.907 µg/day in product P9 (pharmaceutical herbal drug) to 297.320 µg/day reported for product T3 (traditional herbal remedy). The acceptable limits are 730 µg/day for men and 750 µg/day for women in the Mexican population [35,36]. This indicates that all products analyzed were within the recommended limits. The lowest DID for Cu was found for pharmaceutical herbal products when compared to dietary supplements (Table 3; *p =* 0.02), as well as for traditional herbal remedies (Table 3; *p =* 0.011), which showed the highest DID.

The toxic metals studied—Pb, As, and Cd—were analyzed for the products studied. With regards to Pb, the lowest daily intake of 0.049 µg/day was for product P10—a pharmaceutical herbal product—while the highest daily intake of 20.467 µg/day was for a traditional herbal remedy (product T2). The upper limit of Pb exceeded that allowed by both the American Herbal Product Association (AHPA) [28] for the T2 product (traditional herbal remedy), and by US California Proposition 65 (US CA P65) [30] for products S6, T1, and T2, which are a dietary supplement and two traditional herbal remedies, respectively. Nevertheless, the lowest DID was obtained in pharmaceutical herbal products, compared to dietary supplements (Table 3; *p =* 0.003), of which 8% of products exceeded the recommended limits of DID indicated by the AHPA [28]. Moreover, the DID for pharmaceutical herbal products was lower than for traditional herbal remedies (Table 3; *p =* 0.011), in which 100% of products exceeded the DID values recommended by the AHPA [28].

The DID for As was calculated for 14 products, with a range of 0.045–18.624 µg/day, with the lowest value for a pharmaceutical herbal product (P3) and the highest value for a traditional herbal remedy (T3). Product T3 (traditional herbal remedy) exceeded the daily intake values reported by the AHPA [28], US CA P65 [30], and Canada Natural and Non-Prescription Health Products (CNNHP) [29], which report a maximum of 10 µg/day. In particular, the lowest DID was found in pharmaceutical herbal products when compared to dietary supplements (Table 3; *p =* 0.021). For traditional herbal remedies 33% of products exceeded the DID value reported by the AHPA [28].

The DID values for Cd were between 0.0008 and 0.334 µg/day, with the lowest amount for product P10 (pharmaceutical herbal product) and the highest for product S6 (dietary supplement). All products were within the acceptable values for this toxic metal, since the highest amount was lower than the recommended limit of 4.1–6.0 µg/day according to the AHPA, US CA P 65, and the CNNHP [28,29,30].

Table 3 shows that DIDs for pharmaceutical herbal products had the following order Mn > Cu > Pb > As > Cd. For dietary supplements, this pattern was Mn > Cu > Pb > As > Cd, and for traditional herbal remedies it was Cu > Mn > Pb > As > Cd. We found that the average DIDs for each group of items (pharmaceutical herbal products, dietary supplements, and traditional herbal remedies), for each metal analyzed, were lowest for pharmaceutical herbal products, followed by dietary supplements and traditional herbal remedies. All products were contaminated with Pb, 54% of them with As, and 81% of them with Cd. The lowest values of Pb, As, and Cd were detected in pharmaceutical herbal products > dietary supplements > traditional herbal remedies. All pharmaceutical herbal products were within the limits established for Mn, Cu, Pb, As, and Cd. For dietary supplements, the limits for Pb were exceeded for one product, but Mn, Cu, As, and Cd were within the limits. Finally, for traditional herbal remedies, all products exceeded the limits for Pb, but Mn, Cu, As, and Cd were all within the limits.

### 3.2. Non-Carcinogenic Health Risk Assessment

#### 3.2.1. Risk Calculation for Human Health

Table 4 shows the EDI values (mg/kg/day) of the different metals analyzed (Mn, Cu, Pb, As, and Cd) in pharmaceutical herbal products, dietary supplements, and traditional herbal remedies.

EDIs for Mn were calculated for the whole range of products, and showed values between 7.090 × 10^−7^ and 2.528 × 10^−4^ mg/kg/day, with the lowest value for product P9 (pharmaceutical herbal product) and the highest for a traditional herbal remedy (product T2). The EDIs that were estimated for Cu in all products had values between 1.396 × 10^−5^ and 2.78 × 10^−3^ mg/kg/day, for products P9 (pharmaceutical herbal product) and T1 (traditional herbal remedy), respectively.

In particular, the EDIs obtained for Pb had a minimum value of 7.523 × 10^−7^ for product P10 (pharmaceutical herbal product) and a maximum value of 1.398 × 10^−4^ for product T3 (traditional herbal remedy). EDIs for the toxic metal As (inorganic) were calculated for 14 products; the values ranged from 6.863 × 10^−7^ mg/kg/day for the lowest value, a pharmaceutical herbal product (P3), to the highest value for product T3 (traditional herbal remedy), with 2.865 × 10^−4^ mg/kg/day.

EDI values for Cd were calculated for 21 products, with a range from 1.231 × 10^−8^ to 4.36 × 10^−6^ mg/kg/day, for products P10 (pharmaceutical herbal product) and S13 (dietary supplement), respectively.

The estimation of the EDI values for the pharmaceutical herbal products presented the following order: Mn > Cu > Pb > As > Cd. For dietary supplements, this pattern was Mn > Cu > Pb > As > Cd, while for traditional herbal remedies it was Cu > Mn > Pb > As > Cd.

#### 3.2.2. Hazard Quotient Estimation

The calculation of the hazard quotients for Mn, As, and Cd from the consumption of herbal products (pharmaceutical herbal products, dietary supplements, or traditional herbal remedies) is reported in Table 5. The evaluation of risk is the process of analyzing the potential health effects in humans from doses of the pollutant received. All hazard quotients for Mn, As, and Cd were below 1 (HQ < 1). The HQ values for the analyzed metals were highest in traditional herbal remedies, followed by dietary supplements and, finally, the pharmaceutical herbal products.

#### 3.2.3. Hazard Index Estimation

The cumulative non-carcinogenic risk estimation was calculated for Mn, As, and Cd, and called the hazard index (HI). The estimation of HIs for all products analyzed was below 1 (Table 5). For one product (T3; traditional herbal remedy) this value was close to 1.

## 4. Discussion

The World Health Organization (WHO) has reported that around 80% of the world’s population uses medicinal herbs for the treatment of various diseases [4], as occurs in Mexico. Thus, it is important to know if the sale in Mexico of herbal products containing *Ginkgo biloba* is safe in terms of the content of essential and toxic metals, because their high consumption may be detrimental to human health.

In the current study, we found that 100% of the products analyzed (pharmaceutical herbal products, dietary supplements, and traditional herbal remedies) that are marketed in the metropolitan area of Mexico City showed the presence of Mn, Cu, and Pb, while As and Cd were detected in 54 and 81% of products, respectively (Table 2). In particular, pharmaceutical herbal products had the lowest levels of Mn, Cu, Pb, As, and Cd, indicating that they are less contaminated, followed by dietary supplements, while the highest content was found in traditional herbal remedies. These findings are in accordance with previous studies in other parts of the world—for example, the USA, India, China, and Brazil—which have reported the frequent contamination of herbal products with toxic metals such as Pb, As, Cu, and Cd [37,38,39,40,41]. Even in countries with strict sanitary control for these types of products—such as Singapore—contamination with As, Pb, and Cu has been found in 4% of the more than 3300 products analyzed [39].

The analysis of the daily intake of the metals analyzed in *G. biloba* products is of importance, because its consumption is very popular in the Mexican population—mainly in the elderly—and it is ingested for memory loss and concentration. It is important to highlight that this highly vulnerable sector of the population is ~20% of the Mexican population [42], so this study is of public health importance.

We found that all products contain Mn and Cu within the daily limits established for the Mexican population [35,36]. Moreover, all of the products tested were contaminated with toxic metals (Pb, As, and Cd), but none of the pharmaceutical herbal products exceeded the daily intake limits proposed by the AHPA, US CA P65, and CNNHP [28,29,30], whereas some dietary supplements and traditional herbal remedies exceeded the recommended daily limits. We used the proposed limits of these metals contained in products of herbal origin set by the AHPA, US CA P65, and CNNHPD, which are the most recent and widely recognized worldwide.

In the current study, daily Pb intake ranged from 0.049 for product P10 to 20.467 µg/day for product T2. In particular, the products S6 (dietary supplement), T1, T2, and T3 (traditional herbal remedies) exceeded the maximum permissible limit for Pb in herbal products, which according to different guidelines is 6 µg/day for the AHPA [28], 10 µg/day for the CNNHPD [29], and 15 µg/day for US CA P65 [30]. Likewise, we found that the daily intake of all of the herbal pharmaceutical products falls within the minimum limits. Pb is highly toxic, and can be absorbed inorganically through ingestion of food, liquids, or inhalation. The harmful effects of acute Pb exposure on human health through herbal products are associated with hematopoietic toxicity, hepatotoxicity, and gastrointestinal symptoms [18,41]. It is also important to emphasize that Pb is used in various industrial activities, such as the manufacture of fertilizer, and that these industrial activities can have an environmental impact in areas of agricultural production [17,18].

After analyzing the daily intake of As, we found that it was in a range between 0.045 for product P3 and 18.624 µg/day for product T3, and that this latter product (traditional herbal remedy) exceeded the daily levels allowed by various international organizations for herbal products—10 µg/day according to the AHPA [28], CNNHP [29], and U.S. CA P65 [30]. We can also highlight that the average calculation of the daily As intake of pharmaceutical herbal products analyzed is lower than that of the dietary supplements (Table 2). It is also relevant to indicate that chronic intoxication due to consumption of As from medicinal herbs produces leukopenia, sensory neuropathy, anemia, neoplasms, and skin changes [43,44]. This metal has been used in agricultural industries, including the preparation of fungicides, herbicides, and algaecides, among other applications.

In our study, daily Cd intake ranged from 0.0008 for product P10 to 0.334 µg/day for product S6. However, none of the products analyzed exceeded the permitted levels of Cd in herbal products according to the values issued by the AHPA (4.1 µg/day), CNNHPD (6.0 µg/day), or US CA P65 (4.1 µg/day) [28,29,30]. The products with the lowest allowable daily intake estimates were pharmaceutical herbal products, followed by dietary supplements and, lastly, traditional herbal remedies. Cd is frequently detected as a pollutant in industrial areas of the pharmaceutical industry, and in the use of fertilizers, among others [45,46]. This may be a factor contributing to Cd contamination in the products tested in the present study.

In a study carried out in Brazil, the presence of Cd and Pb was reported in *G. biloba* products [47], as reported in the current study, although the concentrations of these metals were not sufficient to cause harm to health. In addition, the leaves of this tree are used in both Brazil and Mexico as a medicine to treat memory loss, brain edema, circulatory problems, and cellular aging.

Similar to our findings, commercial preparations containing *Ginkgo biloba* (tablets, leaves, infusions, extracts, and capsules) sold in Spain were studied and various metals were analyzed, and the estimated daily intakes were also calculated. After analyzing Cd and Pb in 80 herbal products, none exceeded the limits established by the European Pharmacopoeia for Medicinal Plants. Furthermore, the intake of these metals in *G. biloba* products was not found to represent a risk to human health [48]. In our study of 26 products tested, 4 products exceeded the limits for Pb (traditional herbal remedies and dietary supplements) and 3 for Cd (traditional herbal remedies), according to the daily limits reported by the AHPA [28].

In addition, severe contamination by heavy metals such as Cd, Cu, and Pb has been reported in *Ginkgo biloba* leaves and barks collected in the Seoul area of South Korea [49,50]. Therefore, it is important to note that *G. biloba* is capable of accumulating metals that are toxic to human health. In particular, *Ginkgo biloba* leaves and seeds are used as traditional herbal remedies, but different leaf extracts are consumed as pharmaceutical herbal medicines as well as dietary supplements on the international market. It has been reported that there are toxicological effects in experimental studies related to the intake of *Ginkgo biloba* seeds and leaves, considered to be a possible human carcinogen by the International Agency for Research on Cancer [51], which could be related to the poor surveillance of these products, which may contain highly toxic metals such as Pb, As, and Cd, which are toxic and mutagenic at very low concentrations [18].

On the other hand, in a similar study conducted in Canada, toxic metals were analyzed in 121 natural health products (traditional Chinese, Ayurvedic, and marine products), as well as 49 pharmaceutical preparations. Toxic metal contamination was found in many of the products, both in supplements and in pharmaceutical preparations. However, only a small percentage of natural health products exceeded the daily limits for Cd, As, Pb, mercury, and aluminum, and according to US CA P 65 [30], no pharmaceutical product exceeded the daily limits [52]. In addition, in a study carried out in Poland, the content of Cd, Pb, and mercury was analyzed in 41 dietary supplements of herbal origin. It was found that 68% of the products were contaminated with Cd and Pb, as reported in our study, although only a single product exceeded the permissible daily limits of Pb according to Poland´s regulations [53].

Today, herbal products are more accessible to international consumers due to globalization—particularly Chinese remedies and Indian Ayurvedic medicine. However, products from Asia are often contaminated with toxic metals such as Pb, As, and mercury [41,44,54]. It should be noted that although China is the main global supplier of *Ginkgo biloba*, there are few reports on the toxic metal content of products containing *Ginkgo biloba*. It is important to note that Mexico is not a producer of *Ginkgo biloba*, and that the raw material of most of the items analyzed in our study was imported. Most of the *Ginkgo biloba* available on the Mexican market comes from plantations in China, France, and the USA [55]. However, due to high demand on the international market, adulteration of this product has been detected from suppliers in Europe, Asia, and North America, which might have serious health effects [55]. Thus, although it is well known that the use of material of natural origin can have health benefits, side effects have been reported [56]. These could be associated with the quality of the raw material, the incorrect identification of the plant leading to incorrect use, impurities, or contaminants such as toxic metals.

The main sources of contamination from toxic metals in the raw material used to produce herbal medicines are soil, water, and air [57]: (1) through the cultivation of plants in soils contaminated with toxic metals, irrigation with wastewater, and the use of compost and animal manure enriched with toxic metals, raw material can absorb toxic compounds; (2) metal cross-contamination can occur during processing phases (washing, drying, grinding, solvent extraction) and preparation of herbal products; (3) the addition of metals as therapeutic compounds is widely used in Ayurvedic medicine where As, Pb, and Cd are considered to have health benefits [58]; (4) another source of exposure is the transport of products in open-bed trucks, which can allow the transfer of contaminants [57]; (5) the use of medicinal plants following inappropriate instructions, improper preparation, use of excessive doses, or prolonged use can also prove hazardous.

In Mexico, there is little information on the content of metals in herbal products. Thus, the most recent study reported the presence of Cu, Pb, and Cd in dietary supplements of herbal origin that included three products with *G. biloba* [37]. However, the estimate of the daily intake showed little risk to health, as reported in the present study.

Estimation of non-carcinogenic risk in humans was calculated using the hazard quotients (HQs) for Mn, As, and Cd. Table 5 shows that the estimated HQ values for each metal did not exceed the reference value of 1. The HQs for the three types of products showed the following order: traditional herbal remedy > dietary supplement > pharmaceutical herbal product. Our results showed that the contaminant with the highest HQ in the products was As. The non-carcinogenic cumulative hazard estimation index (HI) in humans for Mn, As, and Cd had a value less than 1. However, we can highlight that the HI value for product T3 (traditional herbal remedy) was very close to 1 and, thus, this product has a greater potential to pose a risk to human health. This study shows that herbal pharmaceuticals have better control in reducing metal contamination than dietary supplements, but traditional herbal remedies were the most contaminated.

Nevertheless, our results are inconclusive in showing that there are no health risk associated with the intake of products containing *G. biloba*. This is because it was not possible to calculate the hazard quotient and hazard index estimation of Pb and Cu, since the US EPA does not have an available RfD for inorganic Pb and other Pb compounds, because this metal is a toxic element without threshold, and provision of a RfD is therefore not recommended [59]. In the case of Cu, an RfD has not been evaluated.

The current study had some limitations due to financial constraints. Therefore, we were unable to analyze other toxic metals in order to determine whether contamination with these increases health risk. The acquisition and analysis of more herbal products containing *G. biloba* in regions other than the metropolitan area of Mexico City would also have been desirable.

Our results indicate that the intake of medicinal plants for human health should be carried out as for any regulated medicine, because toxic metal content can cause severe damage to human health, depending on individual´s susceptibility and route of administration, as well as the type of exposure (acute, subacute, or chronic). It is also relevant to note that herbal products that are sold as medicines and dietary supplements in various parts of the world are often unregulated. This allows these products to contain pharmaceutical or non-pharmaceutical products, without therapeutic value but with toxic potential. Heavy metal contamination has been frequently reported, and can lead to heavy metal poisoning in people who have ingested herbal medications, herbal dietary supplements, or traditional herbal remedies [58]. This suggests the importance of applying stricter sanitary regulations when using raw materials of natural origin in the manufacture of items for human consumption in order to avoid damage to health.

In particular, the General Health Law of Mexico has regulated products of herbal origin [60] with surveillance through the Comisión Federal para la Protección contra Riesgos Sanitarios (COFEPRIS) [61], which also includes dietary supplements. However, the law for the latter is not as strict as for herbal medicines. Furthermore, no metal limits have been specified for herbal products. Thus, according to this, herbal medicine requires a health record, which shows its quality, safety, and efficacy both technically and scientifically. On the other hand, traditional herbal remedies cannot be sold for the cure or treatment of any disease. They are freely sold, with proven quality and safety, but their efficacy is based on popular knowledge. Although the regulations in Mexico for these products are new, it is recommended to update Mexican law so as to be able to exercise better surveillance of toxic metals in order to improve product quality before marketing to guarantee efficacy, potency, and safety.

## 5. Conclusions

This study shows that pharmaceutical herbal products that are produced under surveillance are less contaminated with toxic metals such as Mn, Cu, Pb, As, or Cd than dietary supplements and traditional herbal remedies. The daily intake of Pb and As via some herbal products exceeded the established limits. Nevertheless, the hazard quotient estimation and hazard index estimation were calculated only for As, Cd, and Mn, because the US EPA does not have RfDs available for Pb or Cu. Therefore, it is necessary to develop other alternatives to calculate the health risks posed by these metals. It is also important to measure other toxic metals associated with the intake of these products.

## Figures and Tables

**Table 1 ijerph-18-08285-t001:** Characteristics of products investigated containing *Ginkgo biloba*.

Pharmaceutical Herbal Products				
Product Code	Origin *	Declaration on the Label (Main Compounds)	Formulation Presentation	Therapeutic Uses
P1	Mexico	Dry extract of *Ginkgo biloba* leaves standardized to 8.8–10.8 mg flavonol glycosides and 2.16–4.80 mg terpene lactones, 40 mg per capsule	Herbal drug (capsules)	Treatment of memory decline, attention span, retention, and mental concentration
P2	Mexico	*Ginkgo biloba* 6c	Herbal drug (drops)	Memory
P3	Mexico	Dry extract of *Ginkgo biloba* 40 mg standardized to 9.6 mg ginkgo flavone glycosides calculated as quercetin and kaempferol	Herbal drug (tablets)	Cerebral circulatory insufficiency, vertigo, tinnitus, memory impairment, motor disorders, mood disorders, peripheral vascular disorders, intermittent claudication, and neurosensory disorders of vascular origin in ophthalmology and otorhinolaryngology
P4	Mexico	Dry extract of *Ginkgo biloba* (EGb761) 40 mg (standardized to 9.6 mg flavonoid glycosides calculated as quercetin and kaempferol)	Herbal drug (tablets)	Treatment of attention span, mental retention, reduction of recent memory, vasoprotective, and antioxidant
P5	Mexico	Dry extract of Ginkgo biloba EGb 761^®^.…..40 mg (standardized to 9.6 mf flavonoid glycosides calculated as quercetin and kaempferol)	Herbal drug (tablets)	Cerebral circulatory insufficiency and its functional manifestations (vertigo, headache, impairment loss, motor and mood disorders). Sequelae of cerebral vascular accidents, peripheral vascular disorders, and neurosensory disorders of vascular origin (vertigo and tinnitus)
P6	USA	Dry extract of *Ginkgo biloba* leaves (Ginkgo) 50 mg	Herbal drug (tablets)	Improves memory
P7	Mexico	*Ginkgo biloba* 40 mg (dry extract of leaves), glutamic acid (250 mg), calcium phosphate dibasic dihydrate 120.24 mg, ascorbic acid (vitamin c) 20 mg, zinc sulfate monohydrate 13.73 mg, pyridoxine (vitamin B6) 0.533 mg, riboflavin (vitamin B2) 0.340 mg, thiamine (vitamin B1) 0.450 mg, cyanocobalamin (vitamin B12) 0.533 mcg	Herbal drug (tablets)	Helps in attention and concentration loss; improves deficiencies of vitamins and minerals
P8	Spain	*Ginkgo biloba* 7.86 g (dry extract) equivalent to 4.7 mg/g of flavonoids	Herbal drug (pharmaceutical suspension)	To treat recent reduction of memory, attention span, concentration, and vertigo
P9	Mexico	Dry extract of *Ginkgo biloba* to 24% equivalent to 9.6 mg of flavone glycosides	Herbal drug (capsules)	To treat recent reduction of memory, the ability to maintain attention, concentration, retention, and vertigo
P10	Mexico	*Ginkgo biloba* 6c	Herbal drug (drops)	Memory impairment, circulation problems, and vertigo
**Dietary supplements**				
S1	Mexico	*Ginkgo biloba*, *Achillea millefolium*, *Smilax aspera*, *Ammi visnaga*, apple vinegar	Dietary supplement (infusion)	Treatment of varicose veins and hemorrhoids
S2	Mexico	*Ginkgo biloba*, *Centella asiatica*	Dietary supplement (drops)	Alzheimer, vertigo, dizziness, and memory loss
S3	Mexico	Ginkgophyta, *Olea europaea*, *Smilax aspera*, *Ferocactus wislizeni*	Dietary supplement (drops)	Asthma attacks, bronchitis, allergies, urinary incontinence, infantile enuresis, bladder inflammation, vaginal candidiasis, kidney tonic, and sexual stimulant
S4	Mexico	*Ginkgo biloba*, soy lecithin, vitamin E (alpha-tocopherol)	Dietary supplements (capsules)	Recommended for diabetic people who have circulation problems, aid in the treatment of aging, stress, and depression
S5	Mexico	*Ginkgo biloba* L. (powdered leaves)	Dietary supplement (capsules)	To treat blood circulation, chronic headaches, and protect the neuronal and cardiovascular systems from aging
S6	Mexico	*Ginkgo biloba* (L. *Salisburia adiantifolia* Smith)	Dietary supplement (leaves)	Venotonic, neuroprotective, peripheral vasodilator, antiplatelet, anti-hemorrhoidal, and diuretic effects
S7	Mexico	*Ginkgo biloba* (*Ginkgo biloba*, L.), soy lecithin, grape (*Vitis vinifera*, L.), horsetail (*Equisetum robustum*, L.), Gorongoro (*Gorongoro officinalis*)	Dietary supplement (capsules)	To treat blood circulation problems, improve mental function, and treat chronic headaches and vertigo
S8	Mexico	*Ginkgo biloba* L. 40 mg (dry extract of leaf), glutamic acid 250 mg, dibasic calcium phosphate (equivalent to 28 mg of calcium) 120.24 mg, ascorbic acid (vitamin C) 20 mg, zinc sulfate monohydrate (equivalent to 29 mg of zinc) 120.24 mg, riboflavin (vitamin B2) 0.340 mg, pyridoxine hydrochloride (vitamin B6) 0.533 mg, thiamine monohydrate (vitamin B1) 0.450 mg, cyanocobalamin (vitamin B12) 0.533 mcg, polyvinylpyrrolidone K-30, talcum powder, anhydrous lactose	Dietary supplement (tablets)	Strengthening memory, nerves, and treating circulation disorder, facial paralysis, and fatigue
S9	Mexico	Vine (*Vitis vinifera*) leaf powder, olive (*Olea europaea*) leaf powder, Gorongoro (*Gorongoro officinalis*) plant powder. It is sold as *Ginkgo biloba*	Dietary supplement (capsules)	To treat arterial hypertension, and oxygenation in the tissues and the brain. Stimulates the immune system, and reduces cholesterol and triglyceride levels in the blood
S10	Mexico	Ginkgo (*Ginkgo biloba* L.)	Dietary supplement (capsules)	To teat blood circulation, chronic headaches, and aging
S11	Mexico	*Ginkgo biloba*, magnesium stearate	Dietary supplement (tablets)	For the circulatory system
S12	Mexico	Ginkgo (*Ginkgo biloba*), Mikania (*Mikania guaco*), red grape (*Vitis vinifera*), yarrow (*Achillea millefolium*), ahuehuete (*Taxodium mucronatum*)	Dietary supplement (capsules)	Brain oxygenation, improves blood circulation and memory
S13	Mexico	Vine (*Vitis vinifera*) leaf powder, olive (Olea europaea) leaf powder, Gorongoro (*Gorongoro officinalis*) plant powder. Is sold as *Ginkgo biloba*	Dietary supplement (capsules)	To treat arterial hypertension, for oxygenation in the tissues and brain, stimulates the immune system, and reduces cholesterol and triglyceride levels in the blood
**Traditional herbal remedies**				
T1	Mexico	*Ginkgo biloba*	Traditional herbal remedy (leaves)	Memory impairment
T2	Mexico	*Ginkgo biloba*	Traditional herbal remedy (leaves)	For the brain, Alzheimer’s disease, and the circulatory system
T3	Mexico	*Ginkgo biloba*	Traditional herbal remedy (leaves)	Memory impairment and fatigue

* Origin of products is according to what the manufacturer declares, but the origin of natural compounds is unknown.

**Table 2 ijerph-18-08285-t002:** Daily intake dose (µg/day) of different metals in pharmaceutical herbal products, dietary supplements, and traditional herbal remedies containing *Ginkgo biloba*.

Pharmaceutical Herbal Products										
Product Code	Daily Intake (µg/Day) Mn (Mean ± SD)		Daily Intake (µg/Day) Cu (Mean ± SD)		Daily Intake (µg/Day) Pb (Mean ± SD)		Daily Intake (µg/Day) As (Mean ± SD)		Daily Intake (µg/Day) Cd (Mean ± SD)	
P1	3.083	±0.940	4.006	±2.399	0.491	±0.024	N.D.	N.D.	0.029	±0.015
P2	0.195	±0.112	1.476	±0.685	0.127	±0.035	N.D.	N.D.	N.D.	N.D.
P3	1.536	±0.026	1.516	±0.146	0.189	±0.021	0.045	±0.022	N.D.	N.D.
P4	0.655	±0.359	0.921	±0.114	0.188	±0.065	N.D.	N.D.	0.002	±0.000
P5	0.657	±0.552	1.382	±1.445	0.178	±0.005	N.D.	N.D.	0.002	±0.000
P6	14.945	±6.607	7.899	±6.619	0.223	±0.019	N.D.	N.D.	0.146	±0.044
P7	16.434	±12.275	2.213	±0.828	0.249	±0.035	N.D.	N.D.	0.112	±0.034
P8	9.911	±1.342	4.657	±3.531	0.139	±0.027	0.216	±0.048	0.006	±0.004
P9	0.046	±0.002	0.907	±0.321	0.104	±0.002	0.090	±0.077	N.D.	N.D.
P10	0.743	±0.256	3.013	±0.701	0.049	±0.052	0.067	±0.092	0.0008	0.0011
**Dietary supplements**										
S1	2.597	±0.110	3.209	±1.015	0.059	±0.072	N.D.	N.D.	0.004	±0.006
S2	63.073	±2.372	8.460	±5.078	0.162	±0.102	N.D.	N.D.	0.007	±0.004
S3	2.541	±2.204	6.590	±1.004	0.344	±0.079	N.D.	N.D.	0.009	±0.005
S4	6.022	±1.310	2.853	±0.925	2.132	±0.108	1.743	±2.253	0.011	±0.011
S5	72.666	±49.342	20.192	±0.210	2.956	±0.953	0.549	±0.598	0.156	±0.060
S6	89.234	±49.999	218.332	±4.277	15.187	±0.191	N.D.	N.D.	0.334	±0.093
S7	59.846	±17.035	20.872	±6.973	2.246	±1.421	0.448	±0.631	0.047	±0.019
S8	13.336	±6.045	2.124	±0.325	0.252	±0.104	0.046	±0.012	0.015	±0.002
S9	127.737	±26.385	18.910	±1.508	3.568	±0.769	1.885	±2.476	0.089	±0.036
S10	59.715	±15.329	7.000	±0.019	1.379	±0.062	0.574	±0.660	0.111	±0.056
S11	93.818	±39.836	15.857	±0.825	2.788	±0.071	0.452	±0.540	0.089	±0.027
S12	192.164	±63.187	14.967	±1.608	2.758	±0.616	0.774	±0.856	0.259	±0.117
S13	166.239	±1.972	43.054	±14.940	2.827	±0.358	0.468	±0.373	0.284	±0.159
**Traditional herbal remedies**										
T1	97.473	±11.260	180.733	±65.929	19.051	±5.132	N.D.	N.D.	N.D.	N.D.
T2	142.665	±51.413	290.539	±115.325	20.467	±12.110	N.D.	N.D.	0.214	0.189
T3	82.960	±11.710	297.320	±44.067	9.084	±9.441	18.624	7.377	N.D.	N.D.
MVL										
AHPA [28]					6 µg/day		10 µg/day ^a^		4.1 µg/day	
CNNHP [29]					10 µg/day		10 µg/day ^b^		6.0 µg/day	
US CA P 65 [30]					15 µg/day		10 µg/day ^a^		4.1 µg/day	
NIN (MEXICO) [35] 2300 µg/day										
Mexican population [36]			730(men), 750(women) µg/day							

MVL: maximum value limit; AHPA: American Herbal Products Association; CNNHP: Canada Natural and Non-Prescription Health Products; US CA P 65: US California Proposition 65; NIN: National Institute of Nutrition (Mexico); N.D.: non-detectable; ^a^: inorganic arsenic; ^b^: total arsenic; Mn: manganese; Cu: copper; Pb: lead; As: arsenic; Cd: cadmium; SD: standard deviation.

**Table 3 ijerph-18-08285-t003:** Differences between groups with respect to metals analyzed.

Daily Intake (µg/Day)	Pharmaceutical Herbal Products			Dietary Supplements			Traditional Herbal Remedies			
Metals	Mean ± SD	Median (25–75 Percentile)	% above MVL	Mean ± SD	Median (25–75 Percentile)	% above MVL	Mean ± SD	Median (25–75 Percentile)	% above MVL	*p*
Mn	4.82 ± 6.43	1.14 (0.65–9.91)	0% ^e^	72.99 ± 61.24	63.07 (13.34–93.82)	0% ^e^	107.67 ± 31.14	97.47 (90.22–120.07)	0% ^e^	0.001 ^a^ 0.01 ^b^
Cu	2.80 ± 2.21	1.86 (1.38–4.01)	0% ^f^	29.42 ± 57.83	14.97 (6.59–20.19)	0% ^f^	256.20 ± 64.44	290.54 (235.64–293.93)	0% ^f^	0.02 ^a^ 0.011 ^b^ 0.013 ^c^
Pb	0.194 ± 0.12	0.18 (0.13–0.22)	0 % ^d^	2.82 ± 3.91	2.25 (0.34–2.83)	7.7 % ^d^	16.20 ± 6.20	19.05 (14.07–19.76)	100% ^d^	0.003 ^a^ 0.011 ^b^ 0.013 ^c^
As	0.104 ± 0.08	0.08 (0.06–0.15)	0 % ^d^	0.77 ± 0.62	0.55 (0.45–0.55)	0 % ^d^	18.62 ƚ	-----	33.3% ^d^	0.021 ^a^
Cd	0.042 ± 0.06	0.006 (0.002–0.07)	0 % ^d^	0.109 ± 0.12	0.089 (0.01–0.15)	0 % ^d^	0.214 ƚ	-----	0% ^d^	

SD: standard deviation; MVL: maximum value limit; AHPA: American Herbal Products Association; NIN: National Institute of Nutrition (Mexico); Mn: manganese; Cu: copper; Pb: lead; As: arsenic; Cd: cadmium; *p*: *p*-value (two-tailed) for Mann–Whitney U test; ^a^: comparison between pharmaceutical herbal products and dietary supplements; ^b^: comparison between pharmaceutical herbal products and traditional herbal remedies; ^c^: comparison between dietary supplements and traditional herbal remedies; ƚ: Detected in a single product; ^d^: percentage of products above MVL reported by AHPA; ^e^: percentage of products above MVL reported by NIN (Mexico); ^f^: percentage of products above MVL for Mexican population [36].

**Table 4 ijerph-18-08285-t004:** Estimated daily intake (mg/kg per day) of different metals in pharmaceutical herbal products, dietary supplements, and traditional herbal remedies containing *Ginkgo biloba*.

Pharmaceutical Herbal Products					
Product Code	EDI (mg/kg/Day) Mn	EDI (mg/kg/Day) Cu	EDI (mg/kg/Day) Pb	EDI (mg/kg/Day) As	EDI (mg/kg/Day) Cd
P1	4.743 × 10^−5^	6.164 × 10^−5^	7.554 × 10^−6^	N.D.	4.415 × 10^−7^
P2	3.004 × 10^−6^	2.271 × 10^− 5^	1.954 × 10^−6^	N.D.	N.D.
P3	2.364 × 10^−5^	2.332 × 10^−5^	2.906 × 10^−6^	6.863 × 10^−7^	N.D.
P4	1.008 × 10^−5^	1.416 × 10^−5^	2.896 × 10^−6^	N.D.	3.593 × 10^−8^
P5	1.010 × 10^−5^	2.126 × 10^−5^	2.734 × 10^−6^	N.D.	2.614 × 10^−8^
P6	2.299 × 10^−4^	1.215 × 10^−4^	3.438 × 10^−6^	N.D.	2.241 × 10^−6^
P7	2.528 × 10^−4^	3.404 × 10^−5^	3.830 × 10^−6^	N.D.	1.726 × 10^−6^
P8	1.525 × 10^−4^	7.165 × 10^−5^	2.140 × 10^−6^	3.319 × 10^−6^	9.231 × 10^−8^
P9	7.090 × 10^−7^	1.396 × 10^−5^	1.593 × 10^−6^	1.379 × 10^−6^	N.D.
P10	1.143 × 10^−5^	4.635 × 10^−5^	7.523 × 10^−7^	1.031 × 10^−6^	1.231 × 10^−8^
**Dietary Supplements**					
**Product Code**	**EDI (mg/kg/Day)** **Mn**	**EDI (mg/kg/Day)** **Cu**	**EDI (mg/kg/Day)** **Pb**	**EDI (mg/kg/Day)** **As**	**EDI (mg/kg/Day)** **Cd**
S1	3.995 × 10^−5^	4.937 × 10^−5^	9.074 × 10^−7^	N.D.	6.795 × 10^−8^
S2	9.703 × 10^−4^	1.301 × 10^−4^	2.492 × 10^−6^	N.D.	1.115 × 10^−7^
S3	3.910 × 10^−5^	1.014 × 10^−4^	5.290 × 10^−6^	N.D.	1.338 × 10^−7^
S4	9.265 × 10^−5^	4.389 × 10^−5^	2.727 × 10^−5^	2.682 × 10^−5^	1.627 × 10^−7^
S5	1.118 × 10^−3^	3.106 × 10^−4^	4.548 × 10^−5^	8.451 × 10^−6^	2.393 × 10^−6^
S6	1.373 × 10^−3^	3.359 × 10^−3^	2.336 × 10^−4^	N.D.	5.145 × 10^−6^
S7	9.207 × 10^−4^	3.211 × 10^−4^	3.456 × 10^−5^	6.892 × 10^−6^	7.209 × 10^−7^
S8	2.052 × 10^−4^	3.268 × 10^−5^	3.870 × 10^−6^	7.004 × 10^−7^	2.257 × 10^−7^
S9	1.965 × 10^−3^	2.909 × 10^−4^	5.489 × 10^−5^	2.901 × 10^−5^	1.363 × 10^−6^
S10	9.187 × 10^−4^	1.077 × 10^−4^	2.122 × 10^−5^	8.837 × 10^−6^	1.702 × 10^−6^
S11	1.443 × 10^−3^	2.440 × 10^−4^	4.289 × 10^−5^	6.956 × 10^−6^	1.364 × 10^−6^
S12	2.956 × 10^−3^	2.303 × 10^−4^	4.242 × 10^−5^	1.191 × 10^−5^	3.991 × 10^−6^
S13	2.558 × 10^−3^	6.624 × 10^−4^	4.350 × 10^−5^	7.200 × 10^−6^	4.362 × 10^−6^
**Traditional Herbal Remedies**					
**Product Code**	**EDI (mg/kg/Day)** **Mn**	**EDI (mg/kg/Day)** **Cu**	**EDI (mg/kg/Day)** **Pb**	**EDI (mg/kg/Day)** **As**	**EDI (mg/kg/Day)** **Cd**
T1	1.500 × 10^−3^	2.781 × 10^−3^	2.931 × 10^−4^	N.D.	N.D.
T2	2.195 × 10^−3^	4.470 × 10^−3^	3.149 × 10^−4^	N.D.	3.297 × 10^−6^
T3	1.276 × 10^−3^	4.574 × 10^−3^	1.398 × 10^−4^	2.865 × 10^−4^	N.D.

EDI: estimated daily intake; Mn: manganese; Cu: copper; Pb: lead; As: arsenic; Cd: cadmium; N.D.: Non-detectable.

**Table 5 ijerph-18-08285-t005:** Hazard quotient and hazard index estimation of different metals in pharmaceutical herbal products, dietary supplements, and traditional herbal remedies containing *Ginkgo biloba*.

Pharmaceutical Herbal Products				
Product Code	HQ Mn	HQ As	HQ Cd	HI
P1	0.000339	N.A.	0.000883	0.001222
P2	0.000021	N.A.	N.A.	0.000021
P3	0.000169	0.002288	N.A.	0.002456
P4	0.000072	N.A.	0.000072	0.000144
P5	0.000072	N.A.	0.000052	0.000124
P6	0.001642	N.A.	0.004482	0.006124
P7	0.001806	N.A.	0.003453	0.005259
P8	0.001089	0.011064	0.000185	0.012338
P9	0.000005	0.004597	N.A.	0.004602
P10	0.000082	0.003437	0.000025	0.003543
**Dietary supplements**				
S1	0.000285	N.A.	0.000136	0.000421
S2	0.006931	N.A.	0.000223	0.007154
S3	0.000279	N.A.	0.000268	0.000547
S4	0.000662	0.089384	0.000325	0.090371
S5	0.007985	0.028170	0.004785	0.040941
S6	0.009806	N.A.	0.010289	0.020095
S7	0.006577	0.022972	0.001442	0.030991
S8	0.001465	0.002335	0.000451	0.004251
S9	0.014037	0.096685	0.002726	0.113448
S10	0.006562	0.029457	0.003404	0.039423
S11	0.010310	0.023186	0.002727	0.036223
S12	0.021117	0.039714	0.007982	0.068813
S13	0.018268	0.023999	0.008723	0.050990
**Traditional herbal remedies**				
T1	0.010711	N.A.	N.A.	0.010711
T2	0.015677	N.A.	0.006594	0.022272
T3	0.009116	0.955077	N.A.	0.964193

HQ: hazard quotient; HI: hazard index; N.A: not applicable; Mn: manganese; Cu: copper; Pb: lead; As: arsenic; Cd: cadmium.

## Data Availability

The data supporting the current study are available upon reasonable request from the corresponding author.

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
