# Peer review of "A Health Risk Assessment of Lead and Other Metals in Pharmaceutical Herbal Products and Dietary Supplements Containing Ginkgo biloba in the Mexico City Metropolitan Area"

_ijerph, 2021, doi:10.3390/ijerph18168285_

Round 1
Reviewer 1 Report
Page 2 – line 65 – 67
"however, due to their high demand"
It is unclear how high demand is related to heavy metal pollution?
Page 2 – line 77 – what is meant by "heavy metals as active compounds"
If the extract is regulated as herbal medicine in Mexico – is heavy metal testing required as part of regulations?
No need to present results at the end of the Introduction section
Results
Provide data on concentrations of heavy metals (and not daily intake)
Results, line 238:
Perhaps more accurate to describe Pb, As, and Cd as toxic – not as "non-essential"
Methods
Why is there no available RfD for lead? Is it possible that this is because there is no quantifiable threshold level?
Conclusions
The statement "the evaluation of the non-carcinogenic….there is no health risk…" is misleading since you were not able to calculate this for lead. If Pb and As exceeded established limits, how can you conclude that there is no health risk? Are established limits based on criteria other than health risk?
Your recommendation is to "strengthen the regulations and surveillance for quality control" – this is very vague – can you provide more specific recommendations?
Author Response
Please see the attachment for reviewer 1

Reviewer 2 Report
Line 69 What about the legacy from paint in soil as manifested by elevated lead levels near the structures of homes etc?
Obeng-Gyasi, E., Roostaei, J. and Gibson, J.M., 2021. Lead Distribution in Urban Soil in a Medium-Sized City: Household-Scale Analysis. Environmental Science & Technology, 55(6), pp.3696-3705.
Line 81: Cadmium is also associated with cardiovascular disease. See:
Tellez-Plaza, M., Guallar, E., Howard, B.V., Umans, J.G., Francesconi, K.A., Goessler, W., Silbergeld, E.K., Devereux, R.B. and Navas-Acien, A., 2013. Cadmium exposure and incident cardiovascular disease. Epidemiology (Cambridge, Mass.), 24(3), p.421
Obeng-Gyasi, E., 2020. Chronic cadmium exposure and cardiovascular disease in adults. Journal of Environmental Science and Health, Part A, 55(6), pp.726-729.
Line 102-105- There is no need to put results in the introduction section.
Line 138: the nitric acid was used to digest…
For the discussion you need to compare your results with other studies more. What new things were learned? Give more context to the results so the reader understands the implications better.
Author Response
Please see the attachment for reviewer 2

Round 2
Reviewer 1 Report
new line 625: "This is because it seems that Pb is a non-threshold toxicant"
I recommend removing "it seems" this is well established.
Also I recommend shortening the discussion section. It is too long and needs to be more focused.
Author Response
"Please see the attachment"
